# POSITIONAL ENCODER GRAPH QUANTILE NEURAL NETWORKS FOR GEOGRAPHIC DATA

## ABSTRACT

Positional Encoder Graph Neural Networks (PE-GNNs) are a leading approach for modeling continuous spatial data. However, they often fail to produce calibrated predictive distributions, limiting their effectiveness for uncertainty quantification. We introduce the Positional Encoder Graph Quantile Neural Network (**PE-GQNN**), a novel method that integrates PE-GNNs, Quantile Neural Networks, and recalibration techniques in a fully nonparametric framework, requiring minimal assumptions about the predictive distributions. We propose a new network architecture that, when combined with a quantile-based loss function, yields accurate and reliable probabilistic models without increasing computational complexity. Our approach provides a flexible, robust framework for conditional density estimation, applicable beyond spatial data contexts. We further introduce a structured method for incorporating a KNN predictor into the model while avoiding data leakage through the GNN layer operation. Experiments on benchmark datasets demonstrate that **PE-GQNN** significantly outperforms existing state-of-the-art methods in both predictive accuracy and uncertainty quantification.

## 1 INTRODUCTION

Large spatial datasets are collected in a wide range of applications in economics (Anselin, 2022), meteorology (Bi et al., 2023), urban transportation (Lv et al., 2014; Derrow-Pinion et al., 2021; Kashyap et al., 2022), social networks (Xu et al., 2020), e-commerce (Sreenivasa & Nirmala, 2019) and other fields. Gaussian Processes (GPs) (Rasmussen & Williams, 2006; Cressie & Wikle, 2011) are a fundamental tool for modelling spatial data on continuous domains. They are flexible and interpretable models for unknown functions, both in spatial and more general regression settings. However, with time complexity $O(n^3)$ and storage complexity $O(n^2)$, naive GP methods quickly become intractable for large datasets. This has led to a large range of approximate inference methods, such as those based on sparse approximations to covariance or precision matrices (Reinhard Furrer & Nychka, 2006; Lindgren et al., 2011), low rank approximations (Cressie et al., 2022) or nearest neighbour approximations (Vecchia, 1998; Datta et al., 2016; Katzfuss & Guinness, 2021).

Given the difficulty of GP computations, it's of interest to explore scalable methods for large spatial datasets using neural networks (NNs) and to enhance their ability to quantify uncertainty. A state-of-the-art method for making spatial predictions using Graph Neural Networks (GNNs) is the Positional Encoder Graph Neural Network (PE-GNN) of Klemmer et al. (2023). Our contribution is to make three key modifications to the PE-GNN architecture to enhance its ability to make accurate spatial predictions and to quantify uncertainty. These modifications will be explained further below.

NNs are popular in data modeling and prediction tasks like computer vision and natural language processing (NLP). However, traditional NNs struggle to handle spatial dynamics or graph-based data effectively. GNNs (Kipf & Welling, 2017; Veličković et al., 2018; Hamilton et al., 2017) offer a powerful and scalable method for applying NNs to graph-structured data. The idea is to share information through the edges of a graph, allowing nodes to exchange information during learning. GNNs are versatile and can uncover nonlinear relationships among inputs, hidden layers, and each node's neighborhood information. The success of GNNs in spatial applications largely depends on the spatial graph construction, including choice of distance metric and the number of neighboring nodes, and traditional GNNs often struggle to model complex spatial relationships. To address

this, Klemmer et al. (2023) introduced the PE-GNN, which enhances predictive performance in spatial interpolation and regression. However, PE-GNN is not designed to provide a full probabilistic description of the target's distribution, and assuming a Gaussian distribution for predictions can lead to poorly calibrated intervals, such as 80% intervals that fail to contain the true outcome 80% of the time. Recently, Bao et al. (2024) proposed a new framework called Spatial Multi-Attention Conditional Neural Processes (SMACNPs) for spatial small sample prediction tasks. SMACNPs use GPs parameterized by NNs to predict the target variable distribution, which enables precise predictions while quantifying the uncertainty of these predictions.

Methods based on quantile regression are an alternative approach to probabilistic forecasting making rapid progress in recent years. Si et al. (2022) introduced a novel architecture for estimating generic quantiles of a conditional distribution, proposing a set of objective functions that lead to enhancements in density estimation tasks. In one dimension, this method produces quantile function regression and cumulative distribution function regression. Kuleshov & Deshpande (2022) argue that the method of Si et al. (2022) is inefficient with high-dimensional predictors. To address this, they modify the original formulation to incorporate a post hoc recalibration procedure whereby an auxiliary model recalibrates the predictions of a trained model. The first model outputs features, usually summary statistics like quantiles, representing a low-dimensional view of the conditional distribution. The auxiliary model, the recalibrator, uses these features as input to produce calibrated predictions using Si *et al.*'s quantile function regression framework. The main drawback is that it requires training two separate models, each needing its own training set.

Our work makes three contributions. (1) We propose a new architecture that merges the two-step procedure of Kuleshov & Deshpande (2022) into a single model by postponing the concatenation of the $\tau$ value proposed by Si et al. (2022). In this way, we enhance the network's ability to model uncertainty and introduce a regularization mechanism. The model becomes robust to high-dimensional predictor spaces, even though few assumptions are made about the form of the target's conditional distribution. This change allows a single model to fully describe the predictive conditional distribution and to generate quantile predictions and prediction intervals as byproducts. It can be applied to any context, not just spatial regression or GNNs. We show how to integrate this strategy into the PE-GNN framework to create an intrinsically calibrated model with no extra computational cost. (2) We introduce a structural change to PE-GNN. Instead of applying the GNN operator to the concatenation of the nodes' features and the spatial embedding, we apply it only to the features. (3) In PE-GNN, the GNN operator uses neighbours' features to create new node representations but does not include the target value of neighboring nodes. Our third contribution introduces the mean target value of a node's neighbours as a feature after the GNN layers, closer to the output. This allows the model to use neighboring observations of the target variable when making predictions.

The structure of this work is as follows: Section 2 offers a brief background overview, Section 3 outlines the proposed method for geographic data prediction, Section 4 shows experimental results on three real-world datasets, and Section 5 concludes.

## 2 BACKGROUND

**Positional Encoder** Inspired by the Transformer architecture (Vaswani et al., 2017) for geographic data (Mai et al., 2020), PE-GNN (Klemmer et al., 2023) employs a PE with two components: a sinusoidal transformation and a fully-connected NN. The first is a deterministic transformation formed by the concatenation of sinusoidal functions, including variations in frequency and scale. The spatial dimensions (typically represented as latitude and longitude) are handled separately. The second component is a fully-connected NN, denoted $NN(\Theta_{PE})$, taking the output produced by the sinusoidal transformation as input and processing it through a fully-connected NN. Let $\boldsymbol{C}_B = [\boldsymbol{c}_1, \ldots, \boldsymbol{c}_{n_B}]^\top$ be the matrix containing the spatial coordinates of a batch of datapoints, typically of dimension $n_B \times 2$, where each $\boldsymbol{c}_i$ corresponds to the pair (latitude$_i$, longitude$_i$). This transformation results in the desired vector space representation, thereby generating the coordinate embedding matrix $\boldsymbol{C}_B^{emb} = PE(\boldsymbol{C}_B, \sigma_{min}, \sigma_{max}, \Theta_{PE}) = NN(ST(\boldsymbol{C}_B, \sigma_{min}, \sigma_{max}), \Theta_{PE})$.

**Graph Neural Network** GNNs are powerful and scalable solutions for representation learning and inference with graph-structured data. They leverage the topological structure of correlations between nearby graph nodes and represent each node in a latent space embedding suitable for the

specific downstream task (Wu et al., 2022). Popular GNN architectures use this graph structure to update the embeddings of each node, considering both the features of each node and its neighbors, in an iterative process (Wu et al., 2022). The first step comprises aggregating features from each node's neighbours. After aggregation, we combine each node's prior representation with the output of the first step. The initial embedding of each node is its feature vector, so $\boldsymbol{H}_B^{(0)} = \boldsymbol{X}_B$. Then, for each GNN layer $k \in \{1, \dots, K\}$, an iteration of the two step process described above is executed.

The most popular GNN architectures follow this backbone, but differ in the way they aggregate neighbours messages and update the embeddings. Graph Convolutional Networks (GCNs) (Kipf & Welling, 2017) are inspired by the convolution operation from Convolutional Neural Networks (CNNs). For weighted graphs, GCN layer $k$ has the following update equation

$$\boldsymbol{H}_B^{(k)} = f^{(k)} \left( \boldsymbol{D}_B^{-1/2} \left[ \boldsymbol{A}_B + \boldsymbol{I}_B \right] \boldsymbol{D}_B^{-1/2} \boldsymbol{H}_B^{(k-1)} \boldsymbol{W}^{(k)} \right), \text{ for } k \in \{1, \dots, K\}. \tag{1}$$

Here, $f^{(k)}$ is an activation function (e.g., ReLU) and $\boldsymbol{W}^{(k)}$ is a matrix of learnable parameters, while the adjacency matrix $\boldsymbol{A}_B$ describes the connectivity of the constructed graph, where unconnected nodes have a value of 0, and connected nodes have an edge weight computed from their distance. $\boldsymbol{D}_B$ is the so-called *degree* matrix and $\boldsymbol{I}_B$ denotes the identity matrix.

**Positional Encoder Graph Neural Network**  Klemmer et al. (2023) proposed a novel approach for applying GNNs to spatial data: PE-GNN. A PE was introduced that takes the set of spatial coordinates for each datapoint as input and produces a vector representing the learned spatial embedding. This vector is then column concatenated with the node features before application of the GNN operator. Thus, for a given batch $B$ of randomly sampled datapoints, the input to the first GNN layer is $\boldsymbol{H}_B^{(0)} = concat\left(\boldsymbol{X}_B, \boldsymbol{C}^{emb}\right)$. PE-GNN also predicts the Local Moran's I (Anselin, 1995) as an auxiliary task (Klemmer & Neill, 2021). Another innovation lies in the training process, as PE-GNN uses a batch-based procedure. At each training step, a random batch $B$ of nodes is sampled, given by $\{p_1, \dots, p_{n_B}\} \in B$. Using only the nodes belonging to the batch, the entire process of constructing the training graph, generating the spatial embedding, column concatenating with the features, and applying the GNN operator is carried out. The loss function used by Klemmer et al. (2023) is given by $\mathcal{L}_B = MSE\left(\hat{\boldsymbol{y}}_B, \boldsymbol{y}_B\right) + \lambda MSE\left(I(\hat{\boldsymbol{y}}_B), I(\boldsymbol{y}_B)\right)$, where $\lambda$ denotes the auxiliary task weight.

**Quantile regression**  Koenker & Bassett Jr (1978) proposed a linear quantile regression model to estimate conditional distribution quantiles. It uses the pinball loss $\rho_\tau(r_i) = \max\left(\tau r_i, (\tau - 1)r_i\right)$, where $r_i = y_i - \hat{q}_i(\tau)$, $\hat{q}_i(\tau) = \boldsymbol{X}_i\hat{\boldsymbol{\beta}}$, and $\tau$ is the desired cumulative probability associated with the predicted quantile $\hat{q}_i(\tau)$. The pinball loss for the $i$-th observation is $\rho_\tau(r_i)$. The loss over a dataset is the average $\rho_\tau(r_i)$ value over all datapoints. A natural extension of quantile linear regression is quantile neural networks (QNNs). This approach is illustrated in Figure 1a, which seeks to estimate the conditional quantiles for a pre-defined grid $(\tau^1, \dots, \tau^d)$. Each quantile is estimated by an independent model (Figure 1a). This can lead to quantile predictions with quantile crossing (e.g., a median prediction lower than the first quartile prediction).

Rodrigues & Pereira (2020) proposed an approach that outputs multiple predictions: one for the expectation and one for each quantile of interest (Figure 1b). The loss function is:

$$\mathcal{L} = \frac{1}{d+1} \left[ MSE\left(\hat{\boldsymbol{y}}, \boldsymbol{y}\right) + \sum_{i=1}^{n} \sum_{j=1}^{d} \frac{\rho_{\tau^j}\left(y_i - \hat{q}_i(\tau^j)\right)}{n} \right]. \tag{2}$$

Si et al. (2022) proposed a method to generate a model that is independent of quantile selection (Figure 1c). For each datapoint sampled during training, $d$ Monte Carlo samples $\tau \sim U(0, 1)$ are drawn. Each sample is concatenated with the datapoint features to obtain a corresponding quantile estimate, so for each datapoint there are $d$ predicted quantiles. The loss function is similar to Eqn. 2, but they predict random quantiles $\mathcal{L} = \frac{1}{n \cdot d} \sum_{i=1}^{n} \sum_{j=1}^{d} \rho_{\tau_i^j}(y_i - \hat{q}_i(\tau_i^j))$. As the network learns, it becomes able to provide a direct estimate to *any* quantile of interest. Hence, this procedure outputs an inherently calibrated model suitable for conditional density estimation. However, Kuleshov & Deshpande (2022) argue this method is inefficient in mid-to-high predictor space dimensions.

Kuleshov & Deshpande (2022) adapted the architecture from Si et al. (2022) into a two-step process for larger predictor spaces (Figure 1d). First, a model is trained to take the original features as inputs

$$\mathcal{L}^1 = \sum_{i=1}^n \frac{\rho_{\tau^1}(y_i - \hat{q}_i(\tau^1))}{n}$$

$$\mathcal{L}^j = \sum_{i=1}^n \frac{\rho_{\tau^j}(y_i - \hat{q}_i(\tau^j))}{n}$$

$$\mathcal{L}^d = \sum_{i=1}^n \frac{\rho_{\tau^d}(y_i - \hat{q}_i(\tau^d))}{n}$$

(a) Non-linear quantile regression using NN.

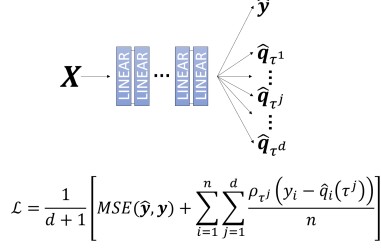

$$\mathcal{L} = \frac{1}{d+1}\left[ MSE(\hat{\boldsymbol{y}}, \boldsymbol{y}) + \sum_{i=1}^n \sum_{j=1}^d \frac{\rho_{\tau^j}\left(y_i - \hat{q}_i(\tau^j)\right)}{n} \right]$$

(b) Non-linear multiple quantile regression.

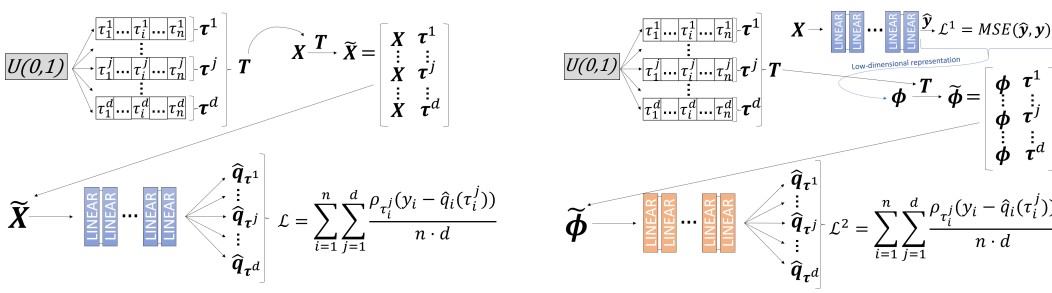

(c) Non-linear quantile function regression.      (d) Two-step density estimation.

Figure 1: (a) For each quantile of interest, a separate NN is trained. (b) Rodrigues & Pereira (2020): one NN outputs $d+1$ predictions: one for the expectation and $d$ for the quantiles. (c) Si et al. (2022): a single NN trained to predict *any* generic quantile of the conditional distribution. (d) Kuleshov & Deshpande (2022): two-step procedure: the first model outputs a low-dimensional representation of the conditional distribution, which a recalibrator then uses to produce calibrated predictions.

and generate low-dimensional representations of the predicted distribution. Next, a recalibrator is trained using these *new* features by minimizing the estimated expected pinball loss over $\tau$. During inference, the recalibrator takes the new features and an arbitrary $\tau$ as inputs to produce the quantile prediction. This method is highly dependent on the choice of recalibrator features.

## 3 METHOD

In this work, we propose a novel approach to spatial data prediction tasks: the Positional Encoder Graph Quantile Neural Network (**PE-GQNN**). Algorithm 1 shows the step-by-step procedure to train a **PE-GQNN** model, and Figure 2 illustrates its complete pipeline. Here, each rectangle labeled "GNN" and "LINEAR" represents a set of one or more neural network layers, with the type of each layer defined by the title inside the rectangle. At each layer, a nonlinear transformation (e.g. ReLU) may be applied. Each datapoint $p_i$ comprises three components $p_i = \{y_i, \boldsymbol{x}_i, \boldsymbol{c}_i\}$. The component $y_i$ is the target variable, and as the focus here is regression, then $y_i$ is a continuous scalar. Additionally, $\boldsymbol{x}_i$ is the feature vector and $\boldsymbol{c}_i$ contains the geographical coordinates associated with observation $i$.

After initializing the model and hyperparameters, the first step of **PE-GQNN** is to randomly sample a batch $B$ of datapoints, $p_1, \ldots, p_{n_B}$. The batch can be fully represented by the target $\boldsymbol{y}_{B\,(n_B \times 1)}$, features $\boldsymbol{X}_{B\,(n_B \times p)}$, and coordinates matrices $\boldsymbol{C}_{B\,(n_B \times 2)}$, respectively. The next step uses the matrix of geographical coordinates $\boldsymbol{C}_B = [\boldsymbol{c}_1, \ldots, \boldsymbol{c}_{n_B}]^\top$ to obtain spatial embeddings for each datapoint (Algorithm 1, Step 5). This process receives $\boldsymbol{C}_B$ as input, and after passing through deterministic sinusoidal transformations and a fully-connected NN, outputs the spatial embedding matrix of the batch $\boldsymbol{C}_B^{emb}{}_{(n_B \times u)}$, containing the spatial context of each pair of coordinates. $\boldsymbol{C}_B$ is also used to compute the distance between each pair of datapoints (Algorithm 1, Step 6). From these distances

---

**Algorithm 1** PE-GQNN training

---

**Require:**
 Training data target, features, and coordinates matrices: $\boldsymbol{y}_{(n\times1)}$, $\boldsymbol{X}_{(n\times p)}$, and $\boldsymbol{C}_{(n\times2)}$.
 A positive integer $k$ defining the number of neighbors considered in the spatial graph.
 Positive integers $tsteps$ and $n_B$, the number of training steps and the batch size.
 Positive integers $u$, $g$, and $s$, the embedding dimensions considered in, respectively, the PE, the
 GNN layers, and the layer where we introduce $\boldsymbol{\tau}$ and $\bar{\boldsymbol{y}}$.
 An activation function $f(\cdot)$ for $\boldsymbol{\tau}$.
**Ensure:**
 A set of learned weights for the model initialized at Step 1.

1: Initialize model with random weights and hyperparameters.
2: Set optimizer with hyperparameters.

3: **for** $b \leftarrow 1$ **to** $tsteps$ **do**                ▷ Batched training
4:  Sample minibatch $B$ of $n_B$ datapoints: $\boldsymbol{X}_{B(n_B\times p)}, \boldsymbol{C}_{B(n_B\times2)}, \boldsymbol{y}_{B(n_B\times1)}$.
5:  Input $\boldsymbol{C}_{B(n_B\times2)}$ into PE, which outputs the batch's spatial embedding matrix $\boldsymbol{C}_B^{emb}{}_{(n_B\times u)}$.

6:  Compute the great-circle distance between each pair of datapoints from $\boldsymbol{C}_B$.
7:  Construct a graph using $k$-nearest neighbors from the distances computed in Step 6.
8:  Set $\boldsymbol{A}_B$ as the adjacency matrix of the graph constructed in Step 7.
9:  **for** $i \leftarrow 1$ **to** $n_B$ **do**
10:   Using $\boldsymbol{A}_B$, compute $\bar{y}_i = \frac{1}{k}\sum_{j=1}^{k} y_j$, where $j = 1,\ldots,k$ are the neighbors of $i$.
11:  **end for**
12:  Set $\bar{\boldsymbol{y}}_B = [\bar{y}_1,\ldots,\bar{y}_{n_B}]^\top$.
13:  Apply GNN layers to the features $\boldsymbol{X}_{B(n_B\times p)}$, followed by fully-connected layers to reduce
  dimensionality. This step outputs a feature embedding matrix $\boldsymbol{X}_B^{emb}{}_{(n_B\times g)}$.
14:  Column concatenate $\boldsymbol{X}_B^{emb}{}_{(n_B\times g)}$ with $\boldsymbol{C}_B^{emb}{}_{(n_B\times u)}$, which results in $\boldsymbol{L}_{B(n_B\times(g+u))}$.
15:  Apply fully-connected layers to reduce $\boldsymbol{L}_{B(n_B\times(g+u))}$ to $\boldsymbol{\phi}_{B(n_B\times s)}$.
16:  Create a vector with values sampled from $U(0,1)$: $\boldsymbol{\tau}_{B(n_B\times1)} = [\tau_1,\ldots,\tau_{n_B}]^\top$.
17:  Column concatenate $\boldsymbol{\phi}_B$ with $f(\boldsymbol{\tau}_B)$ and $\bar{\boldsymbol{y}}_B$ to create $\widetilde{\boldsymbol{\phi}}_{B(n_B\times(s+2))}$.
18:  Predict the target quantile vector $[\hat{q}_1(\tau_1),\ldots,\hat{q}_{n_B}(\tau_{n_B})]^\top$ using $\widetilde{\boldsymbol{\phi}}_B$.
19:  Compute loss $\mathcal{L}_B = \frac{1}{n_B}\sum_{i=1}^{n_B}\rho_{\tau_i}(y_i - \hat{q}_i(\tau_i))$.
20:  Update the parameters of the model using stochastic gradient descent.
21: **end for**

---

and a predefined number of nearest neighbors, a graph can be constructed, with each datapoint as a node and edge weights computed from the distances, leading to the batch adjacency matrix $\boldsymbol{A}_B$.

At Step 13 of Algorithm 1, the first distinction between **PE-GQNN** and PE-GNN arises: instead of using the concatenation of the feature matrix and the spatial embedding as the input for the GNN operator, we apply the GNN operator only to the feature matrix $\boldsymbol{X}_B$. One or more fully-connected layers are then used to reduce the feature embedding dimensionality. This process receives the constructed graph and the batch feature matrix $\boldsymbol{X}_{B(n_B\times p)}$ as input and yields an embedding matrix of features as output: $\boldsymbol{X}_B^{emb}{}_{(n_B\times g)}$. This modification applies the GNN operators exclusively to the features, without smoothing out the PEs. The GNN layers can be of any desired type. Step 14 of Algorithm 1 performs a column concatenation between the feature embedding $\boldsymbol{X}_B^{emb}{}_{(n_B\times g)}$ and the output obtained from the PE: $\boldsymbol{C}_B^{emb}{}_{(n_B\times u)}$ (Figure 2). This concatenation results in the matrix $\boldsymbol{L}_{B(n_B\times(g+u))}$. After that, the other innovations of **PE-GQNN** come into play.

First, we use one or more fully-connected layers (Algorithm 1, Step 15) to reduce the dimensionality of $\boldsymbol{L}_B$, making it suitable for two of the three innovations in **PE-GQNN**. This set of fully-connected layers outputs the matrix $\boldsymbol{\phi}_{B(n_B\times s)}$, which is then combined with $\bar{\boldsymbol{y}}_B$ and $\boldsymbol{\tau}_B$. $\bar{\boldsymbol{y}}_B$ represents a vector with one scalar for each datapoint in the batch, containing the mean target variable among the training neighbours for each node. It is computed using the graph constructed in previous steps

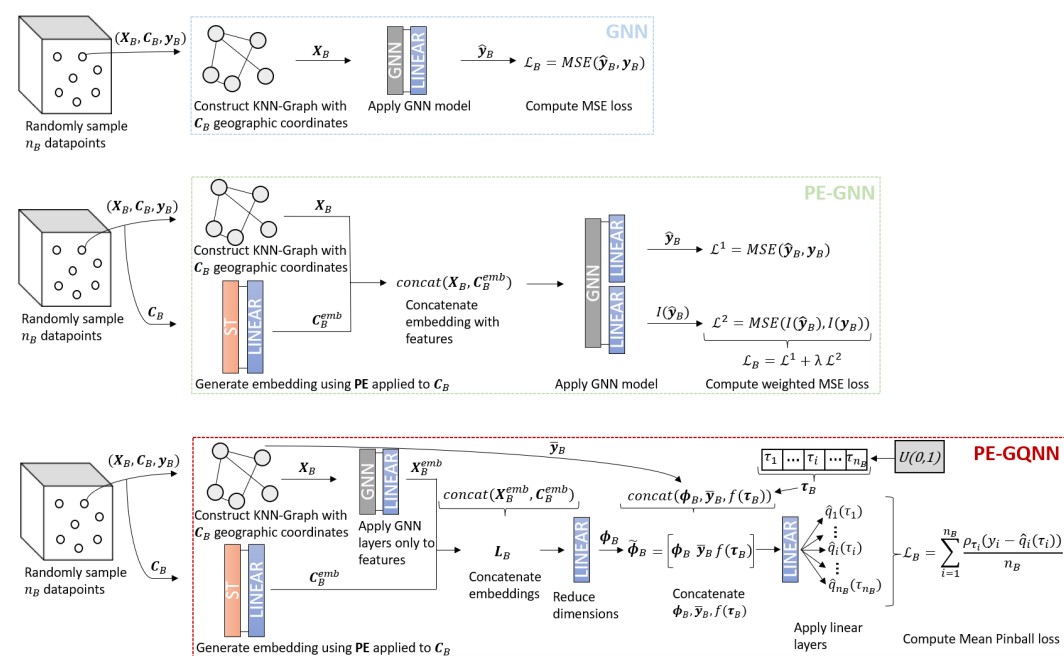

Figure 2: **PE-GQNN** compared to PE-GNN and GNN.

(Algorithm 1, Step 10), and has dimensions $n_B \times 1$. It is comparable to a vector of predictions generated by a KNN regression model, where neighbours are determined using the distance calculated from geographical coordinates. Here, we used the simple average due to its relationship with KNN prediction; however, one could use a weighted average via the adjacency matrix $A_B$. We introduce this input at a later stage to avoid data leakage. If the GNN operator received $\bar{y}_B$ as input, after completing the message passing process in each GNN layer, the true node target value would inadvertently be transmitted to its neighbours, creating potential data leakage (Appleby et al., 2020).

In the same layer where $\bar{y}_B$ is introduced, we apply a similar approach to Si et al. (2022) to make **PE-GQNN** an inherently calibrated model suitable for probabilistic and quantile predictions. For each batch $B$, we create a $n_B \times 1$ vector $\boldsymbol{\tau}_{B(n_B \times 1)} = [\tau_1, \ldots, \tau_{n_B}]^\top$ of random $U(0, 1)$ draws (Algorithm 1, Step 16). Then, we column concatenate $\boldsymbol{\phi}_B$ with $f(\boldsymbol{\tau}_B)$ and $\bar{y}_B$ to create $\widetilde{\boldsymbol{\phi}}_{B(n_B \times (s+2))}$ (Algorithm 1, Step 17), where $f(\cdot)$ is an activation function. Here we propose use of $f(\cdot) = logit(\cdot)$, to facilitate the network's learning. Subsequently, forward propagation is computed (Algorithm 1, Step 18) in one or more fully-connected layers, outputting predicted quantiles for each datapoint in the batch. The batch loss is the one proposed by Si et al. (2022), but with $d = 1$ for the $\tau$ values.

This procedure aims to improve the model's ability to learn the conditional probability distribution of the target variable, enhancing uncertainty estimation and quantile predictions. Instead of introducing $\tau$ values alongside features at the network's input, as suggested by Si et al. (2022), we delay their entry into a reduced latent dimension to boost learning. This adjustment makes **PE-GQNN** suitable for both low- and high-dimensional predictor spaces. It also improves on the Kuleshov & Deshpande (2022) approach by merging the two-network process into a single, intrinsically calibrated model.

Incorporating $\tau$ values into the model architecture improves its ability to model uncertainty and serves as a regularization mechanism (Rodrigues & Pereira, 2020). The use of pinball loss for quantile regression acts as a natural regularizer, producing a detailed description of the predictive density beyond just mean and variance estimation. For predictions, the quantile of interest, $\tau$, must be given, along with the basic data components (e.g. $\tau = 0.25$ gives the first quartile). If interest is in predicting multiple quantiles for the same observation, the input can be propagated up to the layer where $\boldsymbol{\tau}$ is introduced. For each quantile of interest, propagation can be limited to the final layers.

**Target domain**   The final layer should use an activation function coherent with the domain of the target variable, ensuring model outputs are valid for target distribution support. E.g., an exponential function could be appropriate if the target variable is continuous, unbounded and positive.

**Quantile crossing**   This phenomenon occurs when estimated quantile functions for different quantile levels ($\tau$) intersect, violating the requirement that higher quantiles be greater than or equal to lower quantiles. In **PE-GQNN**, by utilizing the same latent representation up to the layer where the quantile level ($\tau$) is introduced, the architecture adopts a hard-parameter sharing multi-task learning framework. This severely mitigates the problem of quantile crossings by constraining the flexibility of independent quantile regression neural network models. If $\tau$ is introduced at the prediction layer, it is guaranteed that quantile crossings will be absent, as the layer equation would be

$$\hat{q}_i(\tau) = f\left(bias + w_\tau \tau + w_{\bar{y}_i}\bar{y}_i + \sum_{j=1}^{neurons} w_j u_j\right), \ \forall i \in 1, \dots, n_B. \tag{3}$$

Here, $neurons$ denotes the number of neurons in the prediction layer, excluding $\tau$ and $\bar{y}_i$. $bias$, $w_\tau$, $w_{\bar{y}_i}$, and $\{w_j\}$ are the prediction layer parameters, and $\{u_j\}$ are the activation values from the previous layer. Commonly, $f$ is chosen to be monotonic, resulting in a monotonic relationship between $\tau$ and $\hat{q}_i(\tau)$. When $\tau$ is introduced at a layer proximal to, but preceding, the prediction layer, the results in Section 4 suggest our approach is not prone to suffer from quantile crossing.

**Number of Monte Carlo samples**   When applying the framework proposed by Si et al. (2022), we chose to use $d = 1$ for the $\tau$ values. Let $\mathcal{L}(\theta, \tau, \boldsymbol{x}, y)$ be the loss function for a given quantile $\tau \sim$ U(0, 1) and an observed pair $(\boldsymbol{x}, y) \sim D_{\text{data}}$, where $D_{\text{data}}$ denotes the full data generative process. On each training iteration, we minimize $\mathcal{L}_B$, which, by the Law of Large Numbers, converges to $\tilde{\mathcal{L}}(\theta) = \mathbb{E}_{\tau, \boldsymbol{x}, y} \mathcal{L}(\theta, \tau, \boldsymbol{x}, y)$, as the batch size, $n_B$, goes to infinity. Therefore, the gradients converge to the same value for any $d$, provided that $n_B \to \infty$. This choice ($d = 1$) simplifies the implementation without sacrificing performance, as shown in Section 4.

## 4 EXPERIMENTS

### 4.1 EXPERIMENTAL SETUP

**PE-GQNN** was implemented using PyTorch (Paszke et al., 2019) and PyTorch Geometric (Fey & Lenssen, 2019). We conducted comprehensive simulations to explore the prediction performance and other properties of the proposed model, comparing it with state-of-the-art methods. Computation was performed on an Intel i7-7500U processor with 16 GB of RAM, running Windows 10.

**Candidate models**   The experiment was designed to compare six primary approaches for addressing spatial regression problems across three distinct real-world datasets. Table 1 lists each candidate model and their applicable datasets. All models were trained using the Adam optimizer (Kingma & Ba, 2015), early stopping and, for all GNN-based models, $k = 5$ nearest-neighbours.

Table 1: Summary of candidate models.

| Approach | Model | Type | Components | | | | | Loss | Datasets |
|---|---|---|---|---|---|---|---|---|---|
| | | | PE | Moran's I | $\tau$ | Structure | $\bar{y}$ | | |
| I | GNN | GNN | No | No | No | No | No | MSE | All |
| II | PE-GNN $\lambda$ = best | GNN | Yes | Yes | No | No | No | $\text{MSE}_y + \lambda\text{MSE}_{I(y)}$ | All |
| III | PE-GQNN $\tau$ | GNN | Yes | No | Yes | No | No | Pinball | California |
| IV | PE-GQNN $\tau$, Structure | GNN | Yes | No | Yes | Yes | No | Pinball | California |
| V | PE-GQNN | GNN | Yes | No | Yes | Yes | Yes | Pinball | All |
| VI | SMACNP | GP | No | No | No | No | No | Log Likelihood | All but 3D road |

Approach I involves the traditional application of GNNs to geographic data. Three types of GNN layers were considered: GCNs (Kipf & Welling, 2017), GATs (Veličković et al., 2018), and GSAGE (Hamilton et al., 2017). For each of these, the architecture remains consistent to facilitate performance comparisons: two GCN/GAT/GSAGE layers with ReLU activation and dropout, followed by a linear prediction layer.

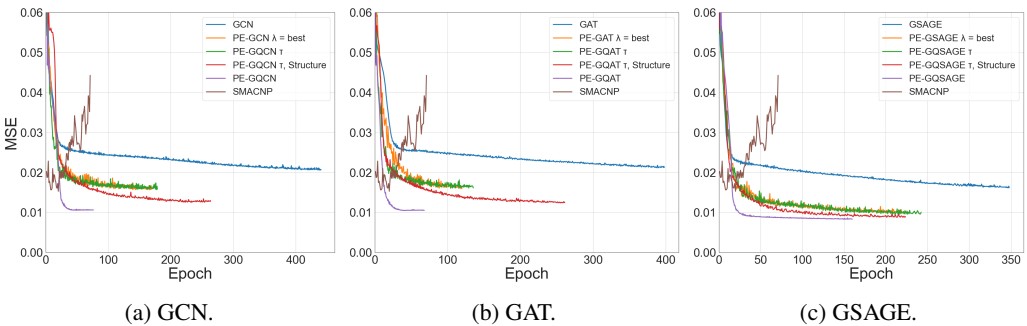

(a) GCN.             (b) GAT.             (c) GSAGE.

Figure 3: Validation error curves on the California Housing dataset, measured by the MSE metric.

Approach II involves the application of PE-GNN (Klemmer et al., 2023) with optimal weights for each dataset and layer type combination, as demonstrated by the experimental findings of Klemmer et al. (2023). The GNN architecture used is the same as for approach I. It was implemented using the code available at: `https://github.com/konstantinklemmer/pe-gnn`.

Approach III is similar to PE-GNN but augmented with the first innovation proposed in this study: the quantile regression framework described in Section 3 is applied. Approach IV is similar to III, but augmented with an additional innovation: the structural alteration in the model's architecture, where the GNN operator is applied only to the features. Approach V, which is the primary focus of this research, explores the utilization of **PE-GQNN**. The PE and GNN layers' architectures remain identical to the previous approaches, with any alterations limited to the proposed innovations.

Finally, a benchmark approach that does not use GNNs but was recently proposed for modelling spatial data will be considered as approach VI: SMACNPs. This approach, proposed by Bao et al. (2024), has demonstrated superior predictive performance, surpassing GPs models in the three real-world datasets considered. This model was implemented following the specifications of Bao et al. (2024), using the code available at: `https://github.com/bll744958765/SMACNP`.

Approaches I and II do not inherently provide predicted conditional distributions. However, as they optimize the MSE metric, they implicitly learn a Maximum Likelihood Estimate (MLE) of a Gaussian model. Thus, the predictive distribution considered for these approaches was a Gaussian distribution centered on the point prediction with variance equal to the MSE of the validation set. For computational simplicity in the experiments, instead of calculating $\bar{y}_B$ for each batch, we pre-calculated $\bar{y}$ using the entire training set.

**Performance metrics** We evaluate predictive accuracy using Mean Squared Error (MSE) and Mean Absolute Error (MAE). To assess calibration of the predictive distributions, we use Mean Pinball Error, $\text{MPE} = \frac{1}{n} \sum_{i=1}^{n} \rho_{\tau_i} (y_i - \hat{q}_i(\tau_i))$, where $\tau_i \sim U(0,1)$, and the Mean Absolute Distance of the Empirical Cumulative Probability, $\text{MADECP} = \frac{1}{99} \sum_{j=1}^{99} \left| \tau^j - \frac{1}{n} \sum_{i=1}^{n} \mathbb{1} \left[ y_i \leq \hat{q}_i(\tau^j) \right] \right|$. For quantile predictions of a calibrated model for a given $\tau$, the proportion of observed values less than or equal to the predicted quantile should approximate $\tau$. Evaluating the MADECP helps determine whether the predicted quantiles are accurate and consistent across the entire space.

## 4.2 CALIFORNIA HOUSING

This dataset comprises pricing information for >20,000 residential properties in California, recorded during the 1990 U.S. census (Pace & Barry, 1997). The main objective is a regression task: predict housing prices, $y$, through the incorporation of six predictive features, $x$, and geographical coordinates, $c$. The predictive features are neighborhood income, house age, number of rooms, number of bedrooms, occupancy and population. All models were trained and evaluated using 80% of the data for training, 10% for validation, and 10% for testing. In the case of SMACNP, to adhere to the specifications of Bao et al. (2024), a training subsample was extracted to represent 10% of the entire dataset. The validation MSE curves throughout training are shown in Figure 3. The number of training epochs and final test dataset performance metrics are in Table 2.

Table 2: Performance metrics on the California Housing test set.

| Model | Epochs | Parameters | MSE | MAE | MPE | MADECP |
|---|---|---|---|---|---|---|
| GCN | 441 | **1,313** | 0.0222 | 0.1101 | 0.0403 | 0.0475 |
| PE-GCN $\lambda = $ best | 170 | 24,129 | 0.0179 | 0.0935 | 0.0354 | 0.0450 |
| PE-GQCN $\tau$ | 179 | 25,217 | 0.0179 | 0.0914 | 0.0351 | 0.0428 |
| PE-GQCN $\tau$, Structure | 264 | 26,169 | 0.0138 | 0.0800 | 0.0302 | 0.0287 |
| PE-GQCN | 76 | 26,201 | 0.0114 | 0.0686 | 0.0272 | 0.0262 |
| GAT | 398 | 1,441 | 0.0227 | 0.1099 | 0.0410 | 0.0586 |
| PE-GAT $\lambda = $ best | 120 | 24,290 | 0.0183 | 0.0930 | 0.0352 | 0.0476 |
| PE-GQAT $\tau$ | 136 | 25,345 | 0.0179 | 0.0926 | 0.0355 | 0.0413 |
| PE-GQAT $\tau$, Structure | 261 | 26,297 | 0.0140 | 0.0829 | 0.0312 | **0.0193** |
| PE-GQAT | **68** | 26,329 | 0.0114 | 0.0685 | 0.0268 | 0.0254 |
| GSAGE | 348 | 2,529 | 0.0170 | 0.0945 | 0.0349 | 0.0569 |
| PE-GSAGE $\lambda = $ best | 222 | 27,426 | 0.0114 | 0.0732 | 0.0280 | 0.0464 |
| PE-GQSAGE $\tau$ | 243 | 28,481 | 0.0113 | 0.0686 | 0.0266 | 0.0478 |
| PE-GQSAGE $\tau$, Structure | 224 | 27,385 | 0.0100 | 0.0632 | 0.0248 | 0.0314 |
| PE-GQSAGE | 160 | 27,417 | **0.0089** | **0.0596** | **0.0229** | 0.0288 |
| SMACNP | 70 | 748,482 | 0.0160 | 0.0881 | 0.0466 | 0.1481 |

As shown in Table 2, **PE-GQNN** achieves state-of-the-art performance metrics, with major improvements over traditional GNN, PE-GNN and SMACNP. For the GSAGE layers, PE-GQSAGE achieved the lowest MSE, MAE and MPE. For this type of layer, which gave the best results overall, we still encounter considerable relative improvements from PE-GQSAGE in comparison with PE-GSAGE, with a reduction of 22% in MSE, 19% in MAE, 18% in MPE, and 38% in MADECP. We can also explore in-depth, the contribution of each specific innovation. The $\tau$ innovation, which corresponds to the application of the quantile regression framework proposed by Si et al. (2022), improved the calibration of quantile predictions, reducing MPE and MADECP. The structural innovation, which involves applying the GNN operator only to the features, is instrumental in enhancing prediction performance and improving the calibration, as evidenced by reduced MSE, MAE, MPE and MADECP. Finally, the use of training neighbours' target mean as a feature introduced at one of the last network layers also further improved the model. This last innovation also accelerated convergence during model training, requiring fewer epochs.

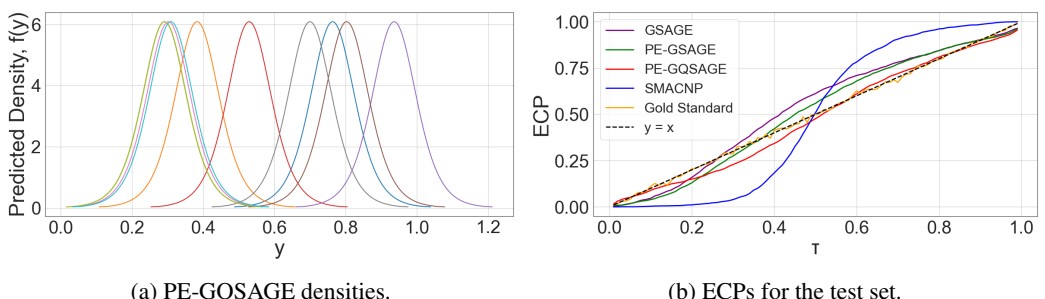

(a) PE-GQSAGE densities.

(b) ECPs for the test set.

Figure 4: (a) PE-GQSAGE predicted densities of 10 observations sampled from the California Housing test set. (b) ECP for each $\tau$ value used for the California Housing test set.

Figure 4 presents plots that elucidate the behavior of the PE-GQSAGE quantile predictions. Figure 4a illustrates the predicted density of a subsample of 10 observations from the test set. For each observation of this sample, the cumulative distribution function was approximated via the quantile predictions using $\tau$ values in $[0.001, 0.002, \ldots, 0.999]^{\top}$. While parametric models presume a rigid structure for their outputs (such as a Gaussian distribution), which constrains their expressiveness, for **PE-GQNN**, no assumptions are made about the form of the predictive distribution. However, as shown in Figure 4a, despite the absence of explicit model restrictions, the model produced symmetric distributional shape across predictions, similar to a Gaussian distribution, in this case.

For all test set observations, we verified that no quantile crossings were observed in any of the **PE-GQNN** models, i.e., all predicted quantiles are monotonically increasing with respect to $\tau$, aligning with the expectations described in Section 3.

Lastly, Figure 4b displays the empirical cumulative probability (ECP) for the test dataset quantile predictions using each of the 99 $\tau$ values in $[0.01, 0.02, \ldots, 0.99]^\top$. This type of plot was proposed by Kuleshov et al. (2018). The closer a model gets to the dashed diagonal line, the closer the $\tau$ values and the ECP. The Gold Standard represents one Monte Carlo draw from a perfectly specified model, where for each quantile level, the ECP is the observed success rate in $n$ Bernoulli trials with a success probability of $\tau$, where $n$ is the number of test set instances. It is evident that PE-GQSAGE has by far the best calibration performance. This is particularly notable when compared to SMACNP, which exhibits substantial calibration deficiencies due to its tendency to overestimate the variance component.

## 4.3 ALL DATASETS

Experiments were conducted on two other geographic datasets used by Klemmer et al. (2023) and Bao et al. (2024). The Air Temperature dataset (Hooker et al. (2018)) contains geographical coordinates for ~3,000 meteorological stations worldwide, with the goal of predicting mean temperatures ($y$) using mean precipitation levels ($x$). Models were trained with 80% of the data, with 10% for validation and testing each, while SMACNP used a 30% subsample for training, following the specifications of Bao et al. (2024). The 3D road dataset (Kaul et al. (2013)), includes $> 430,000$ points with latitude, longitude, and altitude for the Jutland, Denmark road network. The task is to interpolate altitude ($y$) using latitude and longitude ($c$). The data were split into 90% for training, 1% for validation, and 9% for testing. SMACNP metrics are not reported due to high computational costs.

Table 3: Performance metrics from three different real-world datasets.

| Model | California Housing | | | | Air Temperature | | | | 3D road | | | |
|---|---|---|---|---|---|---|---|---|---|---|---|---|
| | MSE | MAE | MPE | MADECP | MSE | MAE | MPE | MADECP | MSE | MAE | MPE | MADECP |
| GCN | 0.0222 | 0.1101 | 0.0403 | 0.0475 | 0.0224 | 0.1158 | 0.0427 | **0.0334** | 0.0170 | 0.1029 | 0.0358 | 0.0560 |
| PE-GCN $\lambda =$ best | 0.0179 | 0.0935 | 0.0354 | 0.0450 | 0.0045 | 0.0467 | 0.0189 | 0.0640 | 0.0032 | 0.0406 | 0.0151 | 0.0476 |
| PE-GQCN | 0.0114 | 0.0686 | 0.0272 | 0.0262 | 0.0025 | 0.0327 | 0.0119 | 0.0713 | **0.0001** | **0.0053** | **0.0022** | 0.0439 |
| GAT | 0.0227 | 0.1099 | 0.0410 | 0.0586 | 0.0233 | 0.1166 | 0.0434 | 0.0497 | 0.0170 | 0.1030 | 0.0359 | 0.0601 |
| PE-GAT $\lambda =$ best | 0.0183 | 0.0930 | 0.0352 | 0.0476 | 0.0058 | 0.0566 | 0.0209 | 0.0960 | 0.0035 | 0.0430 | 0.0163 | 0.0551 |
| PE-GQAT | 0.0114 | 0.0685 | 0.0268 | **0.0254** | 0.0025 | 0.0340 | 0.0143 | 0.0677 | **0.0001** | **0.0053** | **0.0022** | 0.0545 |
| GSAGE | 0.0170 | 0.0945 | 0.0349 | 0.0569 | 0.0223 | 0.1152 | 0.0431 | 0.0361 | 0.0170 | 0.1031 | 0.0358 | 0.0582 |
| PE-GSAGE $\lambda =$ best | 0.0114 | 0.0732 | 0.0280 | 0.0464 | 0.0037 | 0.0449 | 0.0169 | 0.0720 | 0.0032 | 0.0422 | 0.0146 | **0.0417** |
| PE-GQSAGE | **0.0089** | **0.0596** | **0.0229** | 0.0288 | 0.0023 | 0.0326 | 0.0130 | 0.0785 | **0.0001** | 0.0054 | **0.0022** | 0.0786 |
| SMACNP | 0.0160 | 0.0881 | 0.0466 | 0.1481 | **0.0018** | **0.0290** | 0.0391 | 0.2160 | - | - | - | - |

Table 3 showcases the experimental results obtained from all three datasets: California Housing, Air Temperature, and 3D road. Each GNN layer's performance is evaluated across three approaches: the traditional GNN, PE-GNN, and **PE-GQNN**. The **PE-GQNN** models incorporate all three innovations discussed in Section 3. Additionally, we include the SMACNP results as a benchmark model based on GPs. **PE-GQNN** consistently outperforms both traditional GNN and PE-GNN across all datasets and GNN backbones. In every dataset, the **PE-GQNN** innovations lead to significant reductions in MSE, MAE, and MPE. In the California Housing dataset, **PE-GQNN** consistently outperforms SMACNP in predictive accuracy and provides enhanced uncertainty quantification across all types of GNN layers. Conversely, for the Air Temperature dataset, SMACNP achieves the lowest MSE and MAE but suffers from significantly uncalibrated predictions, reflected by a much higher MPE and MADECP compared to **PE-GQNN**.

## 5 CONCLUSION

In this work, we have proposed the Positional Encoder Graph Quantile Neural Network (**PE-GQNN**) as an innovative framework to enhance predictive modeling for geographic data. Through a series of rigorous experiments on real-world datasets, we have demonstrated the significant advantages of **PE-GQNN** over competitive methods. The empirical results underscored the capability of **PE-GQNN** to achieve lower MSE, MAE, and MPE across all datasets and GNN backbones compared to traditional GNN and PE-GNN. Notably, **PE-GQNN** demonstrated substantial improvements in predictive accuracy and uncertainty quantification, as evidenced by its consistent performance in quantile calibration metrics such as MPE and MADECP. The **PE-GQNN** framework's ability to provide a full description of the predictive conditional distribution, including quantile predictions and prediction intervals, provides a notable improvement in geospatial machine learning. **PE-GQNN** provides a solid foundation for future advancements in the field of geospatial machine learning.

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
