# OpenReview forum: "Positional Encoder Graph Quantile Neural Networks for Geographic Data"
_ICLR.cc/2025/Conference — ICLR 2025 Conference Withdrawn Submission_

### Official Review · Reviewer_eA4F · 2024-10-27

**Soundness:** 3
**Presentation:** 3
**Contribution:** 3
**Rating:** 6
**Confidence:** 4

**Summary:**

In this paper, the authors proposed the Positional Encoder Graph Quantile Neural Network (PE-GQNN) as a new framework to enhance predictive modeling for geographic data.  The major contributions of this paper are listed as the following: The empirical results showed the capability of PEGQNN to achieve lower MSE, MAE, and MPE compared to traditional GNN and PE-GNN. Also, PE-GQNN demonstrated substantial improvements in predictive accuracy and uncertainty quantification.

**Strengths:**

There are several strengths demonstrated in the paper:
1. The paper introduces the Positional Encoder Graph Quantile Neural Network (PE-GQNN), a new approach that integrates PE-GNNs, Quantile Neural Networks, and recalibration techniques in a fully nonparametric framework, requiring minimal assumptions about the predictive distributions.
2. The paper has demonstrated the results on three datasets: California Housing, Air Temperature, and 3Droad with 6 different approaches including the proposed PE-GQNN.

**Weaknesses:**

The weaknesses of this paper are listed as the following:

1.	The innovation of this paper seems incremental. Positional Encoder Graph Quantile Neural Network (PE-GQNN) is just a simple combination of PE-GNN with Quantile regression model.

First, the paper shall illustrate in detail the challenges in the integration process. Normally a good integration will include some short cuts to reduce the total cost while comparing with the cost of simple addition of several algorithms together directly. Please try to add some "novel points" or new ideas to demonstrate your merits in integration.

Second, it is better to add some simple examples to illustrate the integration process and novelty in the process on how to design this new graph. Please illustrate which cost you saved compared the cost that you simply integrate several different phases from literature.

Third, will the new integrated framework achieve higher performance compared simply by adding several phases together? What other advantages do you have for the new framework?


2.	The paper shall demonstrate the complexity change for this new PE-GQNN graph.

Is this new graph a simple integration?
How much is the increase of the total complexity?
Is there any new approach to lower the total complexity?

Please provide a specific complexity analysis comparing PE-GQNN to PE-GNN, including time and space complexity.


3.	For the experiments, the following should be addressed.

First, the paper presented experimental results obtained from three datasets: California Housing, Air Temperature, and 3Droad. It seems that the paper lacks discussion about whether other kinds of datasets are suitable for this new approach. Also, how much the total cost changes to implement this new approach.

Second, please discuss the generalizability of their approach beyond the three datasets used. Also, please address what characteristics of a dataset make it suitable for PE-GQNN.

Third, additionally, it would provide valuable practical insights if the authors can demonstrate a comparison of implementation costs between PE-GQNN and existing methods.

**Questions:**

1.  It is better to add some simple examples to illustrate the integration process and novelty in the process on how to design this new graph.

2. The paper shall demonstrate the complexity change for this new PE-GQNN graph.

**Details Of Ethics Concerns:**

The paper has not ethics concerns founded.

---

### Official Review · Reviewer_HAGQ · 2024-10-28

**Soundness:** 2
**Presentation:** 2
**Contribution:** 1
**Rating:** 3
**Confidence:** 4

**Summary:**

This paper proposes a graph neural network-based method for uncertainty quantification in spatial regression.

**Strengths:**

Proposed some techniques for quantifying uncertainty in spatial regression.
Experiments with three datasets.

**Weaknesses:**

The technical contribution of this paper is limited since the proposed method is a combination of Positional Encoder Graph Neural Networks and quantile regression. The proposed method introduces some techniques, such as the use of response variables y and quantile parameters tau in neural networks. However, the novelty of these techniques is incremental.

The experimental results are not convincing. There have been proposed many quantile regression methods in neural networks; i.e., outputting variance in the last layer, the use of sparse Gaussian processes at the last layer. These methods can be easily combined with graph neural networks. The comparison with such existing methods can demonstrate the effectiveness of the proposed method.

**Questions:**

What does ST stand for in the last sentence of the 1st paragraph of Section 2?

In Figure 4(a), why does the predicted density of PE-GQSAGE look like Gaussian distribution? The proposed method outputs predicted quantiles at the given tau. So, the prediction by the proposed method does not necessarily to be Gaussian shape.

---

### Official Review · Reviewer_ZdXk · 2024-10-30

**Soundness:** 2
**Presentation:** 2
**Contribution:** 2
**Rating:** 1
**Confidence:** 4

**Summary:**

The paper introduced a distribution-free uncertainty quantification framework by integrating PE-GNNs, Quantile Neural Networks for geographic data.

**Strengths:**

1. The framework is easy to understand and follow.
2. The final experimental results indicate some effectiveness of the proposed method.

**Weaknesses:**

1. The term "fully nonparametric" could be misleading. It would be more accurate to use "distribution-free."

2. The novelty of the proposed method is limited. Quantile regression is already proposed and widely used in distribution-free uncertainty quantification. For example:
   - "Single-model uncertainties for deep learning." Advances in Neural Information Processing Systems 32 (2019).
   - "Image-to-image regression with distribution-free uncertainty quantification and applications in imaging." International Conference on Machine Learning. PMLR, 2022.

   Additionally, using KNN-Graph with geographic coordinates has been proposed and widely applied in different domains:
   - "Dynamic graph CNN for learning on point clouds." ACM Transactions on Graphics (TOG) 38.5 (2019): 1-12.
   -  "Spatiotemporal graph convolutional networks for earthquake source characterization." Journal of Geophysical Research: Solid Earth 127.11 (2022): e2022JB024401.

3. PE-GNN does not claim the method as SOTA. So which baseline are considered as SOTA in this submission? PE-GNN is better than GNN but not the SOTA. At least there are more available models in applying GNN for earthquake data and traffic data.

**Questions:**

1. The Figure 4(a) does not have legend. What does the 9 curves represent? It is unclear to me how does these 9 curves represent 10 samples.
2. Instead of using MADECP, what is the performance when using popular adopt interval? For example, 95% confidence interval?
3. The real-world geographic data usually collect in a temporal manner, so how does the data is separated into different train/val/test segments? It leads to data leakage issue for sure if the separation does not only use historical data for training.
4. The sensor of the geographic data also will be removed/replaced/re-deployed/defected from time to time in real world applications. Does the proposed method also robust in these cases?

---

### Official Review · Reviewer_x5DA · 2024-10-31

**Soundness:** 3
**Presentation:** 3
**Contribution:** 2
**Rating:** 5
**Confidence:** 3

**Summary:**

This paper presents the Positional Encoder Graph Quantile Neural Network (PE-GQNN), a novel model for spatial data prediction. PE-GQNN integrates Positional Encoder Graph Neural Networks (PE-GNN) with Quantile Neural Networks (QNN) to improve predictive accuracy and quantify uncertainty. Experiments on real-world datasets show that PE-GQNN consistently outperforms existing methods in predictive accuracy and uncertainty calibration.

**Strengths:**

- PE-GQNN combines quantile prediction and distribution recalibration in a single model (not two-stage models), enhancing both predictive accuracy and calibration efficiency.
- By limiting GNN operations to specific features and introducing target values near the output layer, the model effectively prevents data leakage and improves computational efficiency.
- The authors use pinball loss for quantile regression, which allows one to provide a regularization effect, improving prediction reliability across diverse quantile levels.
- Extensive experiments on multiple real-world datasets demonstrate PE-GQNN’s superior performance in both predictive accuracy and uncertainty estimation compared to existing methods.

**Weaknesses:**

- The proposed method is practically useful, but the way PE-GNN and QNN are combined is somewhat straightforward.
- There are several innovations in the architecture, but they are all practical techniques, not theoretically sophisticated.
- There is no discussion of computational cost for high-dimensional data.
- There is no discussion of the shortcomings of the proposed method.

**Questions:**

- Please elaborate on the limitation of the proposed method (e.g., computational cost).
- Is the proposed method robust against noise?

---

### Note · Authors · 2024-11-18

I have read and agree with the venue's withdrawal policy on behalf of myself and my co-authors.